# Counterfactual Vision-Language Data Synthesis with Intra-Sample Contrast Learning

## Abstract

Existing Visual Learning (VL) benchmarks often contain exploitative biases. Most former works only attempted to mitigate biases in semantically low-level and conventional visual-question-answering typed datasets like VQA and GQA. However, these methods cannot generalize to recently emerging highly semantic VL datasets like VCR and are also difficult to scale due to many severe problems like high-cost labors, drastically disrupting the data distribution, *etc.*To resolve those problems and also address other unique biases on VCR-like datasets, we first conduct in-depth analysis and identify important biases in VCR dataset. We further propose a generalized solution that synthesizes counterfactual image and text data based on the original query's semantic focus while producing less distortion to the data distribution. To utilize our synthesized data, we also design an innovative intra-sample contrastive training strategy to assist QA learning in Visual Commonsense Reasoning (VCR). Moreover, our synthesized VL data also serve as a highly-semantic debiased benchmark for evaluating future VL models' robustness. Extensive experiments show that our proposed synthesized data and training strategy improve existing VL models' performances on both the original VCR dataset and our proposed debiased benchmark.

## 1 Introduction

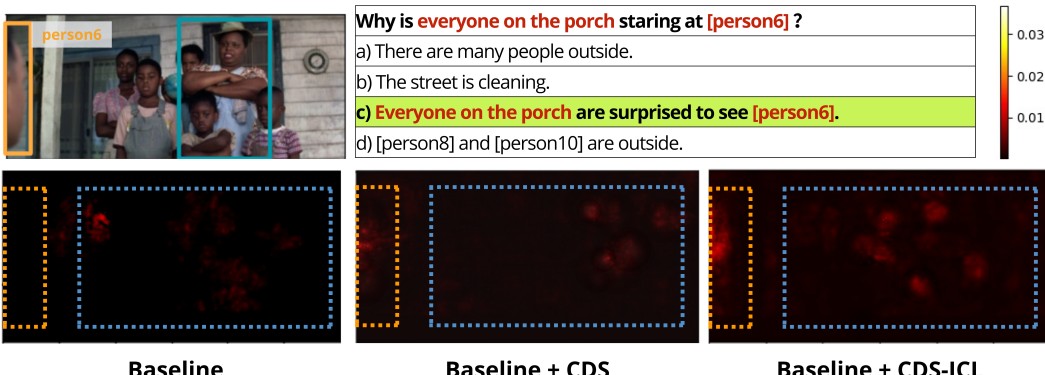

Figure 1: An example from VCR and the paired visual Grad-CAM result from a finetuned $VL - BERT_L$ Su et al. (2019). With training, we expect the VL model to integrate mulitmodal information and commonsense when select the correct answer (labelled by a green check). For instance, to answer this question, we expect the model to focus on [person6] on the left and people in the center. However, the model fails to pick up the correct visual clue and focuses on the irrelevant entity, the window in the background. The orange words are the overlapping words between the correct choice and the question.

Recently many works have explored Vision-Language (VL) models' learning in high semantics from image and text data. As a result, many VQAT-like benchmarks such as GQAHudson & Manning (2019), VQALi et al. (2018), VCRZellers et al. (2019) and SNLI-VEXie et al. (2019) were proposed to evaluate models' abilities in learning visual commonsense and reasoning. Despite recent

successes on aforementioned VL tasks, severe drawbacks due to biases in these evaluation still exist and remain unsolved. These biases prevent existing VL models from understanding the image and text but encourage them to learn shallow mappings based on repeated patterns or spurious distributions Cao et al. (2020); Manjunatha et al. (2019); Ye & Kovashka (2021); Kovaleva et al. (2019); Ramakrishnan et al. (2018); Selvaraju et al. (2020); Agrawal et al. (2018).

Among those datasets, Visual Commonsense Reasoning (VCR) Zellers et al. (2019) has become one of the most recognized highly-semantic VL benchmarks with heavy manual annotations, over the past years. Despite its intention, VCR also suffers from various visual and language biases as illustrated in Fig.1. **Firstly**, the correct choice shares the most number of overlapping words with the question Ye & Kovashka (2021); **Secondly**, the distractors (incorrect choices) are low-quality and often not related to the image and question at all. **Lastly**, as pointed out by previous works, VL models often fail to understand the visual scene and thus maintain very low utilization in visual data Cao et al. (2020); Ye & Kovashka (2021); Agrawal et al. (2018); Ramakrishnan et al. (2018). As shown by the Grad-CAM results Selvaraju et al. (2016) in Fig. 1, models like VL-BERT often focus on regions irrelevant to the question while deducing the correct answer. As a result of these biases in VL datasets like VCR, existing VL models can easily learn shortcuts without utilizing commonsense and reasoning. Therefore, a slight domain-shift in the question, answer or image may drastically affect VL models' predictions Dancette et al. (2021), leading to problems such as prediction inconsistency Selvaraju et al. (2020); Ray et al. (2019); Ribeiro et al. (2019).

Different from conventional datasets like VQA, GQA but similar to other highly semantic ones Tapaswi et al. (2016); Park et al. (2020), VCR has more complex images, diverse question types and follows the standard Multiple-Choice-Question (MCQ) format where each question has multiple sample-specific choices in sentences. Therefore VCR-like dataset may have specific hidden biases unseen before. However, former works only focus on countering biases in the conventional VL datasets which makes them not applicable to VCR. For instance, Agrawal et al. (2018); Dancette et al. (2021)'s methods for balancing answer distribution does not apply to VCR since VCR has question-specific choices in sentences; directly placing occlusions or maskings on image or text Chen et al. (2020a); Liang et al. (2020) may drastically disturb the data distribution in VCR;additional high-cost manual annotations are not practical Ray et al. (2019); Selvaraju et al. (2016); Ribeiro et al. (2019); VCR does not have questions asking only about colors or numbers Gokhale et al. (2020); Inter-sample contrastive learning between image and text pairs may distract models from intra-sample differentiation among VCR's sample specific choices*, etc.*. Chen et al. (2020a); Liang et al. (2020); Gokhale et al. (2020);

Many problems like the above mentioned prevail in former methods and prevent them from generalizing to highly semantic VL datasets like VCR. To raise the community's attention in biases of highly semantic VL datasets like VCR and countering them, in this work, we first conduct in-depth analysis and identify unique biases in VCR. Second, we propose a generalized Counterfactual Vision-Language Data Synthesis (CDS) method to help counter the identified biases. CDS utilizes adversarial models to modify images and answer choices to create synthesized positive and negative image and text data without drastically disturbing data distribution like direct occlusions. Further, we prove that CDS's synthesized data can compliments VCR data to effectively mitigate our identified biases and even integrate to a debiased evaluation benchmark to evaluate future models's robustness.

To better leverage our synthesized data in training, we also propose Intra-sample Contrastive Learning (ICL) framework to assist existing VL models focus on intra-sample differentiation among answer choices and images. Unlike Chen et al. (2020a); Gokhale et al. (2020), ICL frees us from creating paired answers for negative synthesized images. With extensive experiments, we demonstrate that ICL with synthesized data can help existing VL models to be more robust in terms of domain-shifts in data.

In conclusion, our contributions are four-folds.

**Firstly**, we identify significant biases in VCR and analyze VL models' over-reliance on text data.

**Secondly**, we propose an innovative counterfactual VL data synthesis method, CDS, to mitigate the dataset biases. This is the first work to propose an adversarial VL data synthesis method in VCR.

**Thirdly**, to better leverage our synthesized data in training, we further propose an intra-sample contrastive learning mechanism to assist the conventional QA learning with cross entropy loss. To

the best of our knowledge, we are the first to adopt a contrastive learning strategy with counterfactual VL data in highly-semantic VL datasets like VCR and prove its effectiveness.

**Lastly**, our synthesized VL data complements the VCR data and mitigate biases. Therefore, the generated data also serve as a highly semantic VL debiased benchmark on top of VCR for evaluating VL models' performance in a more challenging setting. We also conduct extensive experiments to prove that both our synthesized data and training strategy massively improve VL models' performance on the original VCR and our debiased evaluation benchmarks.

## 2  RELATED WORK

### 2.1  BIASES IN VISION-LANGUAGE DATASETS

Many works have explored biases in VL dataset, especially in the conventional ones like VQA Li et al. (2018) and GQA Hudson & Manning (2019). Since they do not have sample-specific distractors, all questions share the same set of answer choices. Agrawal et al. (2018); Dancette et al. (2021); Zhang et al. (2016) point out that shallow biases exist in the question-answer distribution with such limited variance. Ramakrishnan et al. (2018); Manjunatha et al. (2019) focus on effects of language priors in VQA like shortcuts from the prior words of questions. Due to lazy human annotations and less complexity in text than visual data, VL models can easily learn shortcuts from text biases. Therefore, simple changes in text may lead to significant performance drops and prediction inconsistency Shah et al. (2019); Li et al. (2018); Selvaraju et al. (2020); Ribeiro et al. (2019); Ray et al. (2019). On the other hand, Cao et al. (2020); Wang et al. (2022) also confirms that existing VL models tend to under-utilize visual information compared to text. However, nearly all former works focus on conventional VQA, GQA and other similar VL datasets. Recent highly semantic VL dataset like VCR Zellers et al. (2019); Tapaswi et al. (2016) with MCQ format have answer distribution with larger variance, more diverse question types and complex images. Therefore many of the explored bias analysis may not apply directly. Only one former work, Ye & Kovashka (2021), mentions the bias problem in VCR. However, it suffers from very limited scope when only looking at pronoun words. In this work, we provide an in-depth analysis of biases in an existing highly semantic VL dataset, VCR.

### 2.2  MITIGATING BIASES

Various methods have been proposed to counter biases. Chen et al. (2020a); Liang et al. (2020); Gokhale et al. (2020) place occlusions or maskings on images and questions to create synthesized data. However, occlusions and maskings drastically disturb data distribution while leading to nonsensical synthesized answers. Additionally, these methods do not apply to VCR due to their need for inter-sample contrastive learning. Agrawal et al. (2018); Dancette et al. (2021) categorize answer choices of the whole VQA and rearrange the overall distribution. However, these methods do apply to VL dataset with sample-specific distractors. Ray et al. (2019); Selvaraju et al. (2020); Ribeiro et al. (2019) utilize annotated sub-questions to enhance models' prediction consistency. Nevertheless, these methods have high manual cost and hard to generalize.Gokhale et al. (2020) applies adversarial methods to modify the images but only limit to questions asking about colors and numbers. Other methods Niu et al. (2021); Wang et al. (2022); Zhang et al. (2021b); Niu & Zhang (2021); Gupta et al. (2022) either require largely pretrained VL models, additional data resources or focus only on one modality and hence are hard to generalize. Differently, CDS produces synthesizd positive and negative data for both images and texts. ICL can further provide an effective intra-sample contrastive learning strategy with simple setups while maintaining base models' structures.

## 3  DATASET BIASES AND SHORTCUTS

Former works Cao et al. (2020); Ye & Kovashka (2021); Agrawal et al. (2018); Ramakrishnan et al. (2018) have identified that existing VL models are vulnerable to over-reliance on data from text modality and under-utilization from visual modality. Different from the conventional VL datasets like VQA Antol et al. (2015) and GQA Hudson & Manning (2019), questions and answers in highly semantic VL datasets Zellers et al. (2019); Xie et al. (2019); Park et al. (2020); Tapaswi et al. (2016) contain much more implicit semantic information and images are also more complex to understand.

Hence, it may be more difficult to create those highly semantic dataset without biases and supervise VL models' training on them without picking up shortcuts. To further quantify this issue, we finetune two T5-Base Su et al. (2019) models on the training set of both VQA-Multiple-Choice and VCR with only text data. Comparing the validation results ($21\%$ on VQA-Multiple-Choice and $58\%$ on VCR), we notice that a language-only model can achieve higher performance via only relying on text information (including question and answer choices) on the highly semantic VL dataset, VCR. To our surprise, language-only model can achieve more than $50\%$ much higher than the accuracy of random guessing, $25\%$. This indicates potential learnable strong correlations between only texts and groundtruth labels. The correlation concerns us and may also counter the original objective for evaluating VL models' abilities in visual commonsense reasoning via utilizing both image and text.

In this work, we define "shortcuts" in a more generalized form comparing with Ye & Kovashka (2021): a way of selecting the correct answer by reliance on simple (lexical) patterns, matching repeated references without requiring to fully understand the given image and text. In the following, we share two identified shortcuts based on our analysis.

**Shortcuts of Overlapping Words:** As in Tab. 1, for each question-answer-choices pair in VCR, we first preprocess them via simple tokenization and lemmatization into tokens. Through exact matching, we cumulatively count the number of overlapping words between the question against the correct and the incorrect answer choices respectively. Comparing the top two rows, we discover that the correct answer choices have much higher frequency for containing more overlapping words against the question compared with the incorrect ones. We further experiment to measure the validation accuracy of always selecting the choice with the most overlapping words. To our surprise, in the highly curated VCR dataset costing millions, this simple strategy can already bring us an accuracy of $53.14\%$. For verifying that if it is plausible for existing models to take this shortcut, we further extract a subset from the full VCR data with which the former finetuned language-only T5 model can achieve more than $90\%$ confidence in prediction. After conducting the same analysis on this subset, as shown in the third and fourth rows of Tab. 1, we realize that, within $77.81\%$ of this subset, the correct answer choices have the most overlapping words across all choices. This obviously verifies that existing models are capable of utilizing this shortcut.

**Shortcuts of Low-quality Distractors:** Due to the nature of MCQ, the quality of incorrect answer choices (distractors) also profoundly affect the difficulty of selecting the correct choice. Based on our experiences of dataset annotation, cognitively it is often much more difficult to come up with a quality distractor than the obvious correct choice even for human annotators. Referring Zellers et al. (2019), for every VCR question, the distractors are actually derivation of correct choices of other questions after modification with heuristic rules. With the well-konwn lazy annotation biases, it is not surprising for us to realize that most questions in VCR may be paired with much low-quality distractors. We started with transforming questions and answer choices into a common feature space via a pretrained BERT model Reimers & Gurevych (2019). Then, for every image-question pair: We calculate the cosine similarity of its paired correct answer choice against all the answer choices of other image-question pairs in the dataset and sort them based on the similarity. Similarly, we also calculate the cosine similarity between every correct choice against the three incorrect ones then average them all across samples. We find out that for every sample, the average similarity score between the correct and the incorrect is 0.31. To our surprise, it is even lower than the score between the 1000th ranked similar choice sorted across the dataset and the correct choice. This implies that existing VCR incorrect choices are even outside the top 1000th window of the most similar choices against the correct answer. With applying a simple K-Nearest-Neighbors algorithm Kozma (2008), we further cluster all the answer choices into 20 clusters. For every correct choice, we calculate the similarity score of it against the furthest choice in the same cluster. The average score is 0.34 and it is also higher than the average score between the correct and the incorrect ones for every image-question pair. Regarding this, we believe existing distractors in highly semantic VL dataset like VCR suffer biases of low-quality and this may lead to shortcuts models can easily pick up.

## 4  METHOD

There are two approaches in terms of countering biases and learning shallow shortcuts: the first one is to directly modify the data to mitigate biases in the dataset and the second one is to modify the training strategy of models to avoid learning the shortcuts. In this work, we propose Counterfactual

Vision-Language Data Synthesis and Intra-sample Contrastive Learning (CDS-ICL) framework that tackle those two aspects in a unified way. Given an input image $I$, question $Q$, and answer $A$, we apply CDS to generate positive and negative(counterfactual) synthesized image and answer data, $(I^+, I^-)$ and $(A^+, A^-)$. Unlike Chen et al. (2020a); Liang et al. (2020); Gokhale et al. (2020), Intra-sample Contrastive Learning (ICL) helps bypass the need of creating paired synthesized answers for counterfactual images.

## 4.1 CDS-TEXT-POSITIVE

**External Knowledge:** As discussed in the previous section, correct answers often share more overlapped words and syntactical structures with the question. To mitigate this dataset bias as well as derive more synthesized choices, our strategy is to replace words, phrases of the correct choice with semantic-related alternatives to create variations with fewer overlapped words. Unlike former methods Chen et al. (2020a); Liang et al. (2020); Gokhale et al. (2020) that simply place masking tokens and drastically disturb the original data distribution, our synthesized text data is human-readable with semantically consistent noises. We first utilize some external database to retrieve alternative words to replace words in the correct choice, especially the overlapped ones. The alternative words consist of synonyms and hypernyms from a lexical database, WordNet Fellbaum (2010), and also connected concept-words from ConceptNet Speer et al. (2017) through a set of selected relationships (Referring to Appendix).

**Pretrained Language Model:** To increase the variation in our synthesized choices and avoid naive and biased templates, we further utilize pretrained language models: We inference a T5-Large model Raffel et al. (2020) finetuned on TaPaCo Scherrer (2020) to generate additional paraphrased variations of the correct choices; We also remove stop words from existing correct answers and keep the rest as input of keywords to a pretrained T5-large model finetuned on CommonGen Lin et al. (2019) to generate new sentences.

**Adversarial Filtering:** For ensuring the quality of our created choices and mitigating the biases, lastly, we apply adversarial filterings. First, we calculate the average number of total overlapped words between the distractors and the image-question pair. To mitigate the bias of having more overlapped words in correct answer choices, we choose to reserve generated positive choices that have the same number of overlapped words as the average number of incorrect choices. Lastly, we combine question-correct-answer pairs into statements/captions, $S$ via heuristic rules[1] and calculate the sentence similarity Reimers & Gurevych (2019) between each of the generated choices filtered from the last step against the corresponding statement and only keep the top three as our final synthesized positive answer choices.

## 4.2 CDS-TEXT-COUNTERFACTUAL

**External Knowledge:** For creating counterfactual answer choices and mitigate the biases of low-quality distractors, we generate alternative distractors that are more similar to the correct answers while preserving the semantic difference. Therefore, we utilize both the original correct and incorrect choices to generate counterfactual distractors. If we modify based on the correct choice, we would only replace the target word with antonyms from WordNet and connected concept words from ConceptNet via a selected relationships[2]. If we modify based on the incorrect ones, we also utilize synonyms and hypernyms from WordNet and connected concepts via another larger selected set of relationships (referring to Appendix) in ConceptNet. Following Ye & Kovashka (2021), we always align the pronouns in the generated incorrect choices against the pronouns in the question.

**Pretrained Language Model:** Additionally, we also paraphrase each original incorrect choice with pretrained language models to generate more variations as before.

**Adversarial Filtering:** Lastly, after grammar filtering, we group all the synthesized counterfactual choices derived from both original correct and incorrect choices together to apply filtering. We first calculate the average number of overlapped words between the correct choice and the image-question pair. To mitigate the low-quality distractor bias, we reserve generated counterfactual choices that have the same number of overlapped words as the original correct choice. Then,

---

[1] Details of heuristic rules is listed in Appendix

[2] (DistinctFrom, Antonym)

based on sentence semantic similarity, we rank the generated choices by their maximum similarity measured against any of the original incorrect choices. After selecting the top twenty choices, we re-rank them based on their similarity against the corresponding statement (combination between question ans answer) and finally only reserve the top three. Hence, we expect the final selected generated distractors to be more relevant to the original question-correct-answer pair.

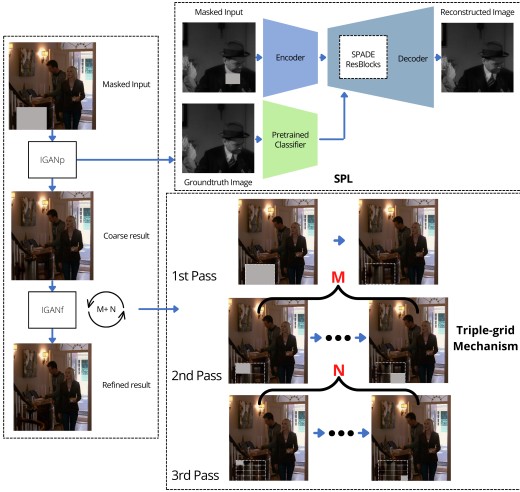

Figure 2: Diagram of Coarse-to-Fine Region Removal.

## 4.3 CDS-IMAGE

Preventing existing models from neglecting visual information and assisting them focus on relevant regions, we propose a region removal method that generates both **I+** and **I-** to emphasize the correct visual dependency. To generate **I+**, we remove image regions that are irrelevant to the question and answer so that the semantic integrity of the image is preserved. On the contrary, we remove relevant image regions to create **I-**. To minimize disturbance in the image distribution, we design a two-stage coarse-to-fine visual in-painting steps.

**Relevant Region Selection:** Many methods have been proposed to select the relevant regions in the image given the question and answer. Das et al. (2017); Selvaraju et al. (2019) use collected human attention maps to select relevant regions on VQA (Antol et al., 2015). However, since human attention maps are often hard to collect, some alternative methods such as Grad-CAM Chen et al. (2020a); Zhang et al. (2021b) and visual-textual co-attention map Lu et al. (2016) are proposed. Nevertheless, as these methods obtain relevant information from the embedding space of a finetuned model, we are concerned that they may already suffer from learning dataset biases based on our analysis before. To ensure correctly determining the relevant entities and regions, we adopt a conditional selection mechanism. At first, we prioritize directly match overlapped tokens between VCR's images and texts via keywords-matching, similar to Gupta et al. (2022). For a $(I, Q, A)$ pair, we construct the image token set $S_I$ by combining the annotated ground-truth object labels and the detected object labels from Anderson et al. (2018). We also further build the textual object set $S_T$ via extracting all the tokens present in the question, the correct answer, and rationale except stop words. We compare every element of $S_I$ against every element of $S_T$. When comparing two objects, we exhaustively apply keywords-matching between them and their corresponding lemmas, stems, and substrings. If there exists any directly matched entity, we prioritize its region as the relevant region. If none is matched, we then apply soft matching via inferencing an off-shelf concept similarity model Zhu & Iglesias (2017) to measure the semantic similarity of every possible pair between the two sets and obtain the scores, $C_{sem}$. Further, we also obtain the statements/captions, $S$ as a combination between $Q$ and $A$ data mentioned before. Afterward, we utilize a pretrained CLIP model Radford et al. (2021) to measure the visual relevance between the region of every entity from $S_I$ and the statements and obtain the scores, $C_v$. Finally, we define relevant regions as: $REL = \{R(s_i) : s_i \in S_I\}$, if $\exists s_t, f(s_i, s_t) = 1$, or $C_{sem}(s_i, s_t) + C_v(R(s_i), S) > T$. $R(s)$ denotes the region for the object $s$ and $f$ represents the keywords-matching procedure. When it equals 1, the two entities are matched. $T$ is a hyperparameter threshold.

**Coarse-to-Fine Region Removal:** After determining the relevant regions in images, following the structure of SPL Zhang et al. (2021a), we design a Coarse-to-Fine In-painting Generative Adversarial Network ($IGAN$) framework to remove relevant and irrelevant regions to generate realistic natural images, **I+/-**. In this framework, we reserve two $IGANs$, an $IGAN_p$ pretrained on Places2 Zhou et al. (2017) as in Zhang et al. (2021a) and another $IGAN_f$ finetuned on VCR. We first retrieve the segmented polygons (if the entity is provided by VCR annotation) or bounding boxes (if generated) of the selected relevant entities. After determining the dimensions of the maximum inscribed rectangles within the polygons or boxes, we create corresponding rectangle maskings in ratios, $(0.7, 0.5, 0.3)$ of the maximum dimensions. Similarly, we also calculate the maximum dimensions of inscribed rectangles in regions that have no entity overlapped on top at all and create maskings of different ratios within those regions. When fine-tuning $IGAN_f$ on the VCR training set, we input images with regions masked by one of those maskings and supervise $IGAN_f$ to reconstruct the masked region. Therefore, essentially $IGAN_f$ is trained to reconstruct the interior of either an entity or an open background region based on its neighboring non-masked pixels and patterns. In order to create **I+**, in inferencing, we filter to irrelevant regions, $R(s_i) \notin$ R E L and create maskings of the minimum circumscribed rectangles around the boxes or polygons. We then feed images masked by those maskings into IGANs to remove the irrelevant entities. Similarly, For creating **I-**, we create similar maskings over the relevant regions from $REL$ to inference to remove the relevant entities. Some examples of the reconstructed images by $IGAN$ are shown in 3.

In order to produce fine-grained images and avoid drastically disturbing existing image distribution and bringing obvious artifacts/biases as occlusion boxes do in Chen et al. (2020a); Liang et al. (2020); Gokhale et al. (2020), in practice, during inferencing, we apply a coarse-to-refine strategy by first passing the masked images into $IGAN_p$ that was pretrained on a larger dataset and then feed the reconstructed output to $IGAN_f$ that was finetuned more specifically to refine the reconstruction. As inferencing in $IGAN_f$, we additionally refine the image via a triple-grid mechanism. As in Fig. 2, for a given masked region, we evenly split it into $M$ blocks and $N$ blocks respectively where $2 < M < N$. In the first pass, we allow $IGAN_f$ to reconstruct the whole masked region in one pass and then revisit the same region with smaller maskings in the following two passes. In the second pass, we take turns to turn each of the $M$ blocks in order into a smaller masking, from the top left to the bottom right, and accordingly reconstruct each masked region to refine. Note that when we reconstruct the first block from the top left of $M$ blocks, the visual regions of the rest $M-1$ blocks are not masked but in-painted with results from the former pass as placeholders. Therefore, we cumulatively inference $M$ times to refine the whole region in the 2nd overall pass. A similar procedure is carried out for the third pass with $N$ blocks. This method allows the framework to maintain the global consistency in reconstructed visual patterns while obtaining the flexibility in refining smaller regions.

### 4.4 CONTRASTIVE LEARNING IN VCR

#### 4.4.1 ANSWER-FOCUSED CONTRASTIVE LEARNING

In former works to resolve Visual-Question-Answering-Type(VQAT) tasks, it is conventional to model as maximizing a probability of answer selection conditioning on the given image and question, $\hat{P}(\boldsymbol{a} \mid I, Q)$. As shown in Fig. 3, it can be essentially regarded as a problem of mapping from $(I, Q)$ to $A$. Hence, former works Chen et al. (2020a); Liang et al. (2020); Gokhale et al. (2020) have to create heuristic methods to create a paired answer for every created counterfactual image. This unfortunately may result in incorrect or counter-intuitive answers with strong biases like "Not ", "Not Green". In this work, due to ICL (discussed later), we do not have to create paired answers for coutnerfactual images. Therefore, we only need to focus on differentiating positive and negative answer choices within each VCR sample. Considering this, we follow the conventional methods to include the Cross Entropy loss of mapping $(I, Q)$ to $A$ via :

$$L_{\text{CE}} = -\sum_{i}^{K} y^i \log\left(\sigma\left(\hat{P}(\boldsymbol{a} \mid I, Q)\right)\right) \tag{1}$$

where $y^i$ is the groundtruth label, $K$ is the total number of samples and $\sigma$ is the softmax function. During QA training, within each batch, we conduct softmax bewteen the logit of each correct answer

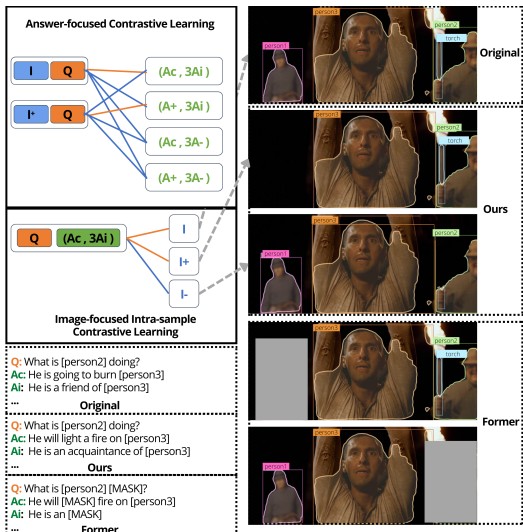

Figure 3: Diagram of all the combinations of (I, Q, A) pairs utilized in training. The pairs form the top block are utilized in QA classification training. The pairs in the bottom block are used in intra-sample contrastive learning. The orange lines indicate the overlapping/repeated pairs. In practice, we only pass those pairs once in the feedforward operations. $Ac$ represents the correct choices. $Ai$ represents the incorrect choices.

and three other incorrect ones. As in Fig. 3, within this training strategy, we can directly augment $I+$ and $A+/-$ into QA training.

### 4.4.2 IMAGE-FOCUSED INTRA-SAMPLE CONTRASTIVE LEARNING

However, it is not straightforward how to apply the counterfactual image, $I-$ into the cross entropy classification loss above. Hence, we decide to transform the problem of mapping $(IQ)$ to $A$, $(IQ \rightarrow A)$ into another related problem of mapping $(QA)$ to $I$, $(QA \rightarrow I)$.

Conventional VQA, GQA datasets do not have sample-specific distractors, and therefore, resolving their tasks, $IQ \rightarrow A$ mapping is essentially learning to differentiate the current sample's answer from other samples' based on the given $(I, Q)$ data. In other words, their tasks focus on inter-sample differentiation. Hence, learning other inter-sample mappings like $QA \rightarrow I$ with contrastive loss (learning to differentiate images of different samples based on the given $(Q, A)$ data) may help resolve the original VQA tasks Li et al. (2018). Differently, VCR has sample-specific distractors. Therefore it focuses more on intra-sample differentiation and learning inter-sample mapping of $QA \rightarrow I$ may not directly help optimize solving the VCR task. On that account, we limit the $QA \rightarrow I$ mapping problem into the intra-sample pairing problem between each $(Q^i, A^i)$ pair against $(I^i, I^i+, I^i-)$ with a loose contrastive loss. As mentioned before, existing VL models under-utilize visual information such that they fail to understand and pick up relevant visual clues from images. Since the contrast among $(I^i, I^i+, I^i-)$ is the presence of relevant visual regions to the question and answer, constraining the learning to it would regulate models to better utilize the visual data in VCR. As shown in Fig. 3, the model would be reinforced to realize the intra-sample visual difference, the relevant entity to the question and answer, the cup. Even though it is not directly optimizing for selecting the answer but it is aiming to select the critical entity the answer mentioned.

We further follow the conventional methods with VL transformer-based models and extract the [CLS] token's hidden feature, $z$ in the end to represent the whole input $(I, Q, A)$ pair, *e.g.* positive pairs like $(I, Q, A)$ and $(I+, Q, A)$ and negative pair $(I-, Q, A)$. Lastly, the contrastive learning is via calculating the InfoNCE loss Van den Oord et al. (2018):

$$\mathcal{L}_{\text{NCE}} = -\log \frac{\exp\left(\Phi\left(\boldsymbol{z}, \boldsymbol{z}_p\right)/\tau\right)}{\exp\left(\Phi\left(\boldsymbol{z}, \boldsymbol{z}_p\right)/\tau\right) + \exp\left(\Phi\left(\boldsymbol{z}, \boldsymbol{z}_n\right)/\tau\right)} \tag{2}$$

where $\Phi$ measures the cosine distance, $\tau$ is a hyperparameter temperature, $z_p$ is the [CLS] token feature for a positive pair, $(I, S)$ or $(I+, S)$, and $z_n$ is for a negative pair, $(I-, S)$.

Finally, the overall objective function is given by:

$$L = \lambda_1 L_{CE} + \lambda_2 L_{NCE} \tag{3}$$

where $\lambda_1$ and $\lambda_2$ are hyperparameter weights.

## 5 EXPERIMENT

In this section, we first describe the datasets including our debiased evaluation benchmark. We then give quantitative analysis on the statistics of biases, ablation, benchmark evaluation and debiased evaluation.

### 5.1 BASE MODEL

CDS-ICL is a generalizable method that can be applied on different VL models. To clearly demonstrate its effectiveness, in this work, we evaluate it with three high-performing baseline methods: one two-stream transformer-based VL model, LXMERT Tan & Bansal (2019) and two one-stream transformer-based VL models, VL-BERT Su et al. (2019) and UNITER Chen et al. (2020b).

### 5.2 DATASET

#### 5.2.1 VCR AND VCR-PRONOUN-SHIFT

We conduct training and evaluation on the train and validation subset of VCR data consisted of around 290K pairs of image, question, and four-way answers. Additionally, for a comprehensive comparison in domain-shift scenarios, we adopt the evaluation with rule-based pronoun-shift from Ye & Kovashka (2021) as one of our evaluation settings, VCR-Pronoun-Shift.

#### 5.2.2 VCR-AS

To fairly compare VL models' capabilities in highly semantic VL data with domain-shift and minimized artifact/noises, we only include the synthesized data samples with synthesized answer data in our own debiased evaluation benchmark. Hence, we include sample data with combinations: $IQ(A_c, 3A_i)$, $IQ(A^+, 3A_i)$, $IQ(A_c, 3A^-)$, $IQ(A^+, 3A^-)$ in our debiased evaluation benchmark with more generalized domain-shift scenarios of answer choices, VCR-Answer-Shift (VCR-AS).

### 5.3 STATISTICS AND BIASES

As in Tab. 1, with CDS-Text, the difference of the average number of overlapping words (against either the VCR question or object labels from the VCR image) between the correct and incorrect choices becomes much less comparing with before. The percentage of incorrect choices that have more overlapping words against the correct ones also increases. Furthermore, after CDS-Text, the averaged semantic similarity between every correct answer choice against the incorrect ones also increases from 0.31 to 0.4 after using CDC-Text.

### 5.4 ABLATION

According the first three rows in Tab. 2, after CDS-Text brings synthesized positive and negative answer choice data and CDS-Image brings synthesized positive image data into the QA classification training with cross entropy loss, the baseline model produces obvious and consistent improved performance. With intra-sample CL, synthesized negative image data is also augmented into the training with contrastive loss and the baseline model can achieve an overall improvement of $1.36\%$. However, after Inter-sample CL, adding images of other samples into the contrastive learning, this distracts the model's attention in intra-sample differentiation especially among answer choices. The severe performance drop confirms with our hypothesis before.

| Choice | VCR Data | CDS-Text | Overlapping Words | | | |
|---|---|---|---|---|---|---|
| | | | w/ Question | | w/ Image | |
| | | | Avg. Num. | More(%) | Avg. Num. | More(%) |
| Incorrect | Train + Val | N | 1.80 | 38.16 | 0.95 | 32.23 |
| Correct | Train + Val | N | 2.02 | 44.44 | 0.97 | 45.75 |
| Incorrect | Language-biased Subset | N | 1.65 | 12.11 | 0.83 | 28.84 |
| Correct | Language-biased Subset | N | 3.14 | 77.81 | 0.98 | 48.16 |
| Incorrect | Train + Val | Y | 1.90 | 40.38 | 0.94 | 35.10 |
| Correct | Train + Val | Y | 1.95 | 42.81 | 0.95 | 42.17 |

Table 1: Average number of overlapped words of correct and incorrect answer choices against the questions and object labels from images.

| CDS-Text | CDS-Image | Intra-sample CL | Inter-Sample CL | VCR |
|---|---|---|---|---|
| | | | | 75.53 |
| ✓ | | | | 75.84 |
| ✓ | ✓ | | | 76.30 |
| ✓ | ✓ | ✓ | | **76.89 (+1.36)** |
| ✓ | ✓ | ✓ | ✓ | 74.16 |

Table 2: Ablation Study. The baseline model is $VLBERT_{Large}$.

## 5.5 Evaluation with VL Benchmark Methods

As Tab. 3, for clearly demonstrating the effectiveness of CDS-ICL, we apply it on top three top-performing VL models on VCR and witness consistent performance boosts.

| Model | Q2A | QA2R | Q2AR |
|---|---|---|---|
| VL-BERT$_L$ $Su et\ al.$ (2019)⋆ | 75.5 | 77.9 | 58.7 |
| VL-BERT$_L$ + $CDS - ICL$ | 76.9 | 78.8 | 60.6 (+1.9) |
| UNITER$_L$ $Chen et\ al.$ (2020b)⋆ | 76.7 | 80.0 | 61.4 |
| UNITER$_L$ + $CDS - ICL$ | 77.3 | 80.8 | 62.5 (+1.1) |
| VILLA$_L$ $Gan et\ al.$ (2020)⋆ | 78.2 | 82.2 | 64.3 |
| VILLA$_L$ + $CDS - ICL$ | 78.7 | 82.6 | 65.0 (+0.7) |

Table 3: Comparing against benchmark methods. ⋆ The results are based on our re-implementation.

## 5.6 Debiased VL Evaluation

CDS-ICL not only can assist existing VL models improve performances in the standard VCR evaluation but also can enhance their robustness against domain-shifts in data. In Tab. 2, with CDS-ICL, base models consistently suffer less performance drops.

| Model | CDS-ICL | VCR | VCR-PS | VCR-AS |
|---|---|---|---|---|
| VL-BERT$_L$ | N | 75.5 | 71.1 | 70.6 |
| | Y | 76.9 | 74.2 | 73.7 |
| UNITER$_L$ | N | 76.7 | 72.8 | 72.0 |
| | Y | 77.3 | 75.4 | 74.7 |

Table 4: Comparing against benchmark methods. ⋆ The results are based on our re-implementation.

## 6 Conclusion

In this paper, we propose a universal multimodal contrastive learning framework with novel data augmentation that jointly learn debiased VL representation for question-answering tasks. Our experiments demonstrate that both the additional generated data and our contrastive learning framework improve models' performance on both the original VCR and our debiased VCR evaluation set while also helping models focus on relevant regions in the image.

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

## A  APPENDIX

You may include other additional sections here.

