# OpenReview forum: "Counterfactual Vision-Language Data Synthesis with Intra-Sample Contrast Learning"
_ICLR.cc/2023/Conference — Submitted to ICLR 2023_

### Official Review · Reviewer_nbLD · 2022-10-21

**Confidence:** 4
**Correctness:** 2
**Technical Novelty And Significance:** 2
**Empirical Novelty And Significance:** 3
**Recommendation:** 3

**Clarity, Quality, Novelty And Reproducibility:**

The paper is hard to read and contains lots of wordy sentences.

The paper has some novelty. The analysis of the overlapped word bias is introduced by Ye & Kovashka(2021). But the paper proposes a new region removal method to obtain counterfactual images.

Minors:

P1L1: Visual Learning -> Visual Language.

Figure 2 and Figure 3 are not centered.

**Strength And Weaknesses:**

Strength:

The paper proposes a new region removal method through impainting instead of masking.

Weakness:

The paper proposes to increase the number of overlapped words in incorrect choices but decrease the number in the correct ones. It may introduce a new bias that the correct answer choices have fewer overlapped words.

Replacing words with antonyms is not guaranteed to generate an incorrect answer. For example, in Figure 1, the changed correct choice "everyone on the porch is unsurprised to see [person6]" still seems a correct answer compared with the other three. The authors do not provide any generated examples to show results are reasonable.

Equation 2 lacks detail. One can generate multiple I+ and thus have more than one positive z_p in equation 2. However, the infoNCE loss can only have one positive z_p and multiple negative z_n. Do the authors randomly select one I or I+ as a positive sample?

The experimental results are weak. For example, the performance of UNITER Large is 77.3%, 80.8%, and 62.8% on the VCR leaderboard. The proposed method does not improve performance.

**Summary Of The Paper:**

The paper focuses on addressing the biases in the VCR dataset. It found that the correct answer has the most overlapping words with the question and the incorrect choices have low bert--similarity with the correct ones. The authors propose to rewrite all answer choices, generate counterfactual images, and train the model with contrastive loss. Firstly, they replace the words with synonyms/antonyms or related concepts, generate new ones with pretrained LM, and filter the results by considering the number of overlapped words and their similarity to the questions. Secondly, the authors generate the counterfactual images by masking and impainting the question relevant/irrelevant regions. Lastly, they minimize the infoNCE loss given the generated positive (I+, Q, A) and negative (I-, Q, A) samples.

**Summary Of The Review:**

Overall, the proposed method neither convince me nor was validated by the experiments. Thus I do not recommend acceptance.

---

### Official Review · Reviewer_EKkA · 2022-10-22

**Confidence:** 4
**Correctness:** 2
**Technical Novelty And Significance:** 2
**Empirical Novelty And Significance:** 2
**Recommendation:** 3

**Clarity, Quality, Novelty And Reproducibility:**

#### **Clarity**

- Although the Abstract and Introduction are compelling, the rest of the parts are not well organized, and the experiments and discussions are not solid enough (see W1).
- Please place figures and tables near the corresponding content. (please check all of them!)
- Notations in Fig. 3 are not defined.
- Notations in Sec 5.2.2 are not defined, e.g., what is the meaning of $3A_i$?
- You did not define VCR-PS, which is presumably VCR-Pronoun-Shift.
- Tasks (Q2A, QA2R, Q2AR) are not defined in Table 3.
- Not `Tab.` but `Tbl.`

#### **Quality**

- In the Abstract, it needs a space after `etc.` before `To resolve`.
- In Sec 3, well-konwn -> well-known
- In Sec 4, a space before `(` in `negative(counterfactual)`
- In Sec 4.3, `are shown in 3` -> `are shown in Fig. 3`?

#### **Novelty**

Although some parts of that are rule-based, generating images and texts for counterfactual samples is an interesting idea to explore. However, it is not free from the inductive bias by the experimenter's design setting aside from increased computational cost.

**Strength And Weaknesses:**

#### **Strength**

For the counterfactual vision-language data synthesis, they carefully generated CDS-Text and CDS-Image for ICL and semantical debiased benchmarks.

#### **Weakness**

- **W1. Evaluation and justification.** The current form of the manuscript incurs many questions. W1-1) How many samples are there in the original VCR and how many counterfactual samples are generated? W1-2) From this, how much it takes to train the model? W1-3) How do we decide this approach is preferable despite of increased training time and computational costs? W1-4) Ablation study for the isolated contribution of CDS would be significant (no ICL, only data augmenting)? W1-5) The improvements in Tables 3 and 4 are statistically significant? W1-6) What was the confidence interval? Therefore, the evaluation and justification of the proposed method are short of expectations for publication.

- **W2. Writing.** There are many minor mistakes in writing and use the notations which are not correctly defined. This makes it hard to follow the content. The paper could be significantly improved by reorganizing and including extensive ablation studies to see the contributions of CDS and ICL and their competitiveness.

**Summary Of The Paper:**

They proposed two methods with different aspects for debiased learning of multimodal learning tasks, e.g., VCR. One is counterfactual vision-language data synthesis (CDS) with linguistic prior knowledge, pretrained lanugage model, and adversarial filtering for textual CDS. And coarse-to-fine In-painting Generative Adversarial Network is used to generate CDS images. While the other is inter-sample contrastive learning exploiting the synthesized dataset. Using the top-performing VL models, they evaluate the proposed method on the VCR benchmarks.


**Summary Of The Review:**

A major part of the paper is allocated to explain the counterfactual vision-language data synthesis; however, since experimental validation is weak, the novelty and significance of this work are incoherent.

---

### Official Review · Reviewer_rKpc · 2022-10-28

**Confidence:** 3
**Clarity, Quality, Novelty And Reproducibility:** Poor.
**Correctness:** 2
**Technical Novelty And Significance:** 3
**Empirical Novelty And Significance:** 3
**Recommendation:** 1

**Strength And Weaknesses:**

Strengths:
1. The findings of the biases in the VCR dataset is inspiring for the entire community. While significant progress has been made and the best accuracy is getting closer to human level, little work has focus on the bias issues (I have to admit I am not an expert in the bias/debias area though. I'll rely on other reviewers' comments to update this claim if it is not correct.).

2. The proposed debias approach shows improved accuracy on the VCR dataset.

Weaknesses:
My major concern is the presentation of the paper. It is not clearly written and hard to follow.
    a) Most importantly, there are no examples of what the altered candidate answers look like. Without them, it is really hard to understand the techniques presented in the paper.
    b) The images in Fig. 2 and 3 are too small. It is hard to clearly see the content.
    c) The usage of the math symbols is a bit arbitrary. For example, $I_+$ and $\textbf{I}_+$ are used in mixture. In the Coarse-to-Fine Regional Removal part, the symbol $REL$ is not defined.
    d) On the bottom of Page 5, it says details of heuristic rules is listed in Appendix, which is totally missing.
    e) It might be a better idea to move Table 1 closer to Section 3 so that the readers can better understand the statistics of the bias in the VCR dataset.

With so many important details missing, I am unable to correctly gauge the technical content of the proposed approach.


**Summary Of The Paper:**

This paper studies the biases in the VCR dataset. It first identifies two important bias problems, by exploiting which significantly better accuracy can be obtained than the random guess. The authors then propose to synthesize counterfactual text and image data to debias the VCR dataset. An intra-sample contrastive learning strategy is used to augment the VQA model training.

**Summary Of The Review:**

The paper should be significantly revised.

---

### Official Review · Reviewer_BL7G · 2022-10-28

**Confidence:** 1
**Correctness:** 4
**Technical Novelty And Significance:** 4
**Empirical Novelty And Significance:** 4
**Recommendation:** 1

**Clarity, Quality, Novelty And Reproducibility:**

The paper exceeds page limits (10 pages versus 9 pages). It should be desk-rejected if we follow the rules.
Thus, my rankings below should not be taken into account.

**Details Of Ethics Concerns:**

The paper exceeds page limits (10 pages versus 9 pages). It should be desk-rejected if we follow the rules.
Thus, my rankings below should not be taken into account.

**Strength And Weaknesses:**

The paper exceeds page limits (10 pages versus 9 pages). It should be desk-rejected if we follow the rules.
Thus, my rankings below should not be taken into account.

**Summary Of The Paper:**

The paper exceeds page limits (10 pages versus 9 pages). It should be desk-rejected if we follow the rules.
Thus, my rankings below should not be taken into account.



**Summary Of The Review:**

The paper exceeds page limits (10 pages versus 9 pages). It should be desk-rejected if we follow the rules.
Thus, my rankings below should not be taken into account.

---

### Decision · Program_Chairs · 2023-01-20

**Decision:**

Reject

**Justification For Why Not Higher Score:**

No reviewers argue for acceptance.

**Justification For Why Not Lower Score:**

N/A

**Metareview: Summary, Strengths And Weaknesses:**

The paper investigates the biases of the VCR dataset, and proposes to cope with these biases by generating counterfactual samples through modifying text and images (e.g. through region removal). While the findings in this work about the biases of VCR methods are important, the paper suffers from clarity issues, and reviewers question aspects of the evaluation.